

# Circular Rosenzweig-Porter random matrix ensemble

Wouter Buijsman⋆ and Yevgeny Bar Lev

Department of Physics, Ben-Gurion University of the Negev,
Beer-Sheva 84105, Israel

⋆ buijsman@post.bgu.ac.il

## Abstract

The Rosenzweig-Porter random matrix ensemble serves as a qualitative phenomenological model for the level statistics and fractality of eigenstates across the many-body localization transition in static systems. We propose a unitary (circular) analogue of this ensemble, which similarly captures the phenomenology of many-body localization in periodically driven (Floquet) systems. We define this ensemble as the outcome of a Dyson Brownian motion process. We show numerical evidence that this ensemble shares some key statistical properties with the Rosenzweig-Porter ensemble for both the eigenvalues and the eigenstates.



## 1 Introduction

Many-body localization is known as a robust mechanism of ergodicity breaking in disordered interacting quantum many-body systems that can be thought of as Anderson localization in Fock space [1, 2]. It is characterized by non-trivial properties, such as the violation of the

eigenstate thermalization hypothesis [3] and the persistent memory of an initial state, among many others [4]. Significant progress towards an understanding of many-body localization from both theoretical and experimental approaches has been witnessed during the last one and a half decade [5–7].

Many-body localization can be mapped to single-particle localization on a Fock space lattice [8], an approach which has been successfully utilized both analytically [9–14] and numerically [15–19]. Within the framework of this approach, obtaining analytical results is notoriously difficult due to strong correlations between disorder potentials [20,21]. It was therefore realized that it is of interest to have simpler phenomenological models for many-body localization that are analytically tractable while still capturing the essential properties [22–24]. A profound example of such a model is the Rosenzweig-Porter random matrix ensemble [25]. This model was suggested as a qualitative phenomenological model for the level statistics and fractality of eigenstates across the many-body localization transition [23], which separates a many-body localized from a thermal phase. This proposal triggered considerable interest for the Rosenzweig-Porter ensemble over the last years [26–35].

The notion of many-body localization extends to interacting periodically driven (Floquet) systems [36–40], for which localization in Fock space has been investigated analytically very recently [41]. While static systems are described by Hamiltonians, which are Hermitian, periodically driven systems are described by Floquet operators, which are unitary. This raises the arguably natural question whether a unitary (or *circular* since the eigenvalues lie on the unit circle in the complex plane) analogue of the Rosenzweig-Porter ensemble can be constructed.

In this work, we propose a circular analogue of the Rosenzweig-Porter ensemble, which we define as the result of a Dyson Brownian motion process [42]. We provide numerical evidence that this circular analogue has similar features as the Rosenzweig-Porter ensemble by investigating some key statistical properties of the eigenvalues and the eigenstates. Our work might serve as a first step in constructing generalizations covering e.g. multifractality, analog to recent physically motivated proposals for generalizations of the Rosenzweig-Porter ensemble [29, 43–46] and other Floquet models with multifractal eigenstates [47–50].

The structure of the paper is as follows. Sec. 2 reviews the Rosenzweig-Porter ensemble and some of its properties. Sec. 3 discusses the construction of the circular Rosenzweig-Porter ensemble. Sec. 4 numerically investigates a number of statistical properties of the circular analogue. Sec. 5 closes the paper with conclusions and an outlook.

## 2 Rosenzweig-Porter ensemble

The Rosenweig-Porter (RP) ensemble [23], which was originally proposed in the context of complex atomic nuclei [25], can be seen as generalization of the Gaussian orthogonal ensemble (GOE) [51,52] with a preferential basis. The ensemble consists of real symmetric matrices $H$ of the form

$$H = H_0 + \frac{\epsilon}{\sqrt{N^\gamma}} V \,, \tag{1}$$

where $N$ is the matrix dimension, $H_0$ is diagonal with elements sampled independently from the Gaussian distribution with mean $\mu = 0$ and variance $\sigma^2 = 1$, and $V$ is sampled from the GOE. The GOE consists of real-valued symmetric matrices with the diagonal and upper triangular elements sampled independently from the Gaussian distribution with mean $\mu = 0$ and variance $\sigma^2 = 2$ (diagonal elements) or $\sigma^2 = 1$ (upper triangular elements). Interpreting $H$ as an Hamiltonian, the parameter $\epsilon \sim \mathcal{O}(1)$ can be viewed of as a perturbation strength. The positive-valued parameter $\gamma$ controls the relative strength of the terms, and thus the properties of the ensemble. Below, we outline a number of these properties, which we numerically explore

for the circular analogue in Sec. 4.

The statistics of spacings between consecutive energy levels provides a standard diagnostic for quantum chaos [3,51]. In the large-$N$ limit, level spacings of the RP ensemble obey Wigner-Dyson level statistics for $\gamma < 2$ [53], which are typically observed for chaotic quantum systems. For $\gamma \geq 2$ and in the same limit, these statistics are Poissonian [53], as typically observed for integrable (non-chaotic) systems. It is convenient to probe level spacing statistics by the average ratio of consecutive level spacings [54,55]. Let $E_n$ denote the eigenvalues of $H$ sorted in ascending order, such that the spacings $s_n$ of consecutive levels are given by $s_n = E_{n+1} - E_n$. The ratios $r_n$ are defined as

$$r_n = \min\left(\frac{s_{n+1}}{s_n}, \frac{s_n}{s_{n+1}}\right). \tag{2}$$

On average, the ratios of consecutive level spacings acquire the values $\overline{r} \approx 0.386$ for Poissonian and $\overline{r} \approx 0.530$ for Wigner-Dyson (GOE) level statistics [55].

Next, we consider eigenstates of $H$ taken from the middle of the spectrum. We focus on the basis in which $H_0$ is diagonal. We denote eigenstates of $H$ and $H_0$ by respectively $|\psi_n\rangle$ and $|n\rangle$. Localization of an eigenstate $|\psi_n\rangle$ can be quantified by the inverse participation ratios

$$\mathrm{IPR}_q = \sum_m |\langle m|\psi_n\rangle|^{2q}\,, \tag{3}$$

with $q > 1/2$. Asymptotically in $N$, the inverse participation ratios scale as $N^{-(q-1)D_q}$, where $D_q$ is known as the fractal dimension with parameter $q$ [30]. Eigenstates of the RP ensemble are characterized by $D_q = 1$ for $\gamma \leq 1$, indicating that these are spread out over a finite fraction of the Hilbert space as $\mathrm{IPR}_2 \sim N^{-1}$. For $1 < \gamma < 2$, the eigenstates are fractal with $D_q = 2 - \gamma$. As $D_q$ does not depend on $q$, the eigenstates are fractal but not multifractal. For $\gamma \geq 2$, the eigenstates are characterized by $D_q = 0$, which reflects that these eigenstates have significant overlap with only $\mathcal{O}(1)$ basis states [23].

Eigenstates of the RP ensemble obey Breit-Wigner statistics [30], which are expected to apply rather generically to quantum many-body systems [56–59]. In the large-$N$ limit, it follows that

$$\overline{|\langle m|\psi_n\rangle|^2} \sim \frac{1}{\left(E_n - E_m^{(0)}\right)^2 + \Gamma^2(E_n)}\,, \tag{4}$$

where the bar denotes an average over $V$ in Eq. (1), $E_n$ is the eigenvalue corresponding to the eigenstate $|\psi_n\rangle$, $E_n^{(0)} = (H_0)_{nn}$, and the so-called spreading width $\Gamma(E_n)$ is obtained through Fermi's golden rule as

$$\Gamma(E_n) = \frac{\pi \epsilon^2}{N^\gamma} \rho(E_n), \tag{5}$$

where $\rho(E_n) = \sum_i \delta(E_n - E_i)$ is the density of states in the vicinity of the eigenvalue $E_n$. Eq. (4) describes what is known as the *shape* of the eigenstates [59]. For $1 < \gamma < 2$, the spreading width gives the width of the so-called mini-band [23,31,60]. In this energy window, the eigenstate amplitudes $|\langle \psi_n|m\rangle|^2$ are on average of order $N^{-D_2}$ (where $D_2$ is the fractal dimension $D_q$ for $q = 2$), which is much larger than the value $N^{-1}$ obtained when averaging over all basis states.

## 3 Circular analogue

In this Section, we propose a unitary analogue of the RP ensemble, which we refer to as the circular RP (CRP) ensemble. The starting point of our construction is the observation made in Ref. [26] that the RP ensemble can be seen as the outcome of a finite-time stochastic process

for the matrix elements, referred to as a Dyson Brownian motion process [42]. Let $M(t)$ denote a time-dependent, real symmetric matrix of dimension $N$, which evolves stochastically according to

$$dM(t) = \sqrt{dt}X, \tag{6}$$

where $dM(t) = M(t+dt) - M(t)$ and $X$ is a GOE matrix sampled independently at each infinitesimal time step $dt$. For the initial condition $M(0) = H_0$ with $H_0$ as given in Eq. (1), one finds $M(t) = H_0 + \sqrt{t}V$, where $V$ is a sample from the GOE. From the solution $M(t)$ is it clear that the RP ensemble results at $t = \epsilon^2/N^\gamma$. The GOE is invariant under transformations of the basis [51, 52]. Using this, for an infinitesimaly small time difference $dt$ one can obtain stochastic equations for the eigenvalues $E_n(t)$ of $M(t)$ perturbatively as

$$dE_n(t) = \sqrt{dt}X_{nn} + dt \sum_{m\neq n} \frac{(X_{nm})^2}{E_n(t) - E_m(t)}, \tag{7}$$

and the stochastic equations for the corresponding eigenstates $|\psi_n\rangle$ as

$$d|\psi_n(t)\rangle = \sqrt{dt} \sum_{m\neq n} \frac{X_{mn}}{E_n(t) - E_m(t)}|\psi_m(t)\rangle - \frac{dt}{2} \sum_{m\neq n} \frac{(X_{mn})^2}{[E_n(t) - E_m(t)]^2}|\psi_n(t)\rangle, \tag{8}$$

where the increments are given by $dE_n(t) = E_n(t+dt) - E_n(t)$ and $d|\psi_n(t)\rangle = |\psi_n(t+dt)\rangle - |\psi_n(t)\rangle$. In the literature, the stochastic evolution of the eigenstates is known as the *eigenvector moment flow* [61, 62].

To construct a circular ensemble, we follow an approach introduced by Dyson [42]. We introduce $S(t)$, which is a time-dependent symmetric unitary matrix of dimension $N$ with eigenvalues $e^{i\theta_n(t)}$. Such a matrix can be interpreted as the Floquet operator of a periodically driven system obeying time-reversal symmetry. The matrix evolves stochastically according to

$$dS(t) = i\sqrt{dt}U^T(t)XU(t), \tag{9}$$

where $dS(t) = S(t+dt) - S(t)$, and $U(t)$ is a unitary matrix defined via the decomposition $S(t) = U^T(t)U(t)$. The decomposition is defined up to transformations $U \to OU$ for real orthogonal matrices $O$[1], which can be absorbed in $X$ as the GOE is invariant under basis transformations. It is useful to take $U^T = U$, in which case $U(t)$ becomes $\sqrt{S(t)}$. Eq. (9) conserves the unitarity of $S(t)$ only in the limit $dt \to 0$. Other, numerically equivalent methods which conserve unitarity of $S(t)$ for any $dt$ can be also used (see Sec. 4). We propose the CRP ensemble as the outcome of the stochastic process described by Eq. (9) at $t = \epsilon^2/N^\gamma$. For the RP ensemble, $M(0)$ is diagonal with uncorrelated elements. This motivates us to analogously take

$$S(0) = \text{diag}\left(e^{i\theta_1(0)}, \dots, e^{i\theta_N(0)}\right), \tag{10}$$

with the phases $\theta_n(0)$ sampled independently from the uniform distribution ranging over $[-\pi, \pi]$. This non-unique choice respects the uniform density of states as observed generically for Floquet operators. The phases evolve according to a stochastic equation that can be obtained as

$$d\theta_n(t) = \sqrt{dt}X_{nn} + dt \sum_{m\neq n} \frac{(X_{nm})^2}{2\tan\frac{1}{2}[\theta_n(t) - \theta_m(t)]}, \tag{11}$$

where $d\theta_n(t) = \theta_n(t+dt) - \theta_n(t)$.

---

[1] The set $U_S(N)$ of symmetric unitary matrices of dimension $N$ is isomorphic to $U(N)/O(N)$, where $U(N)$ denotes the set of imaginary unitary matrices of dimension $N$, and $O(N)$ denote the set of real orthogonal matrices of dimension $N$. For a proof, see e.g. Proposition 2.2.4 of Ref. [52].

Eqs. (7) and (11) are known to yield the same statistics in the middle of the spectrum and in the large-$N$ limit, provided that the densities of states of $M(t)$ and $S(t)$ are the same [63]. Heuristically, this can be understood by noticing that the summations are dominated by contributions from the few most nearby levels, which are of the order of the density of states, $\rho(E_n) \sim N$. While there are $\mathcal{O}(N)$ additional terms in these summations, at the middle of the spectrum the contributions from levels $E_m > E_n$ ($\theta_m > \theta_n$) and $E_m < E_n$ ($\theta_m < \theta_n$) are respectively negative and positive, and therefore provide a contribution with an overall sub-leading magnitude of $\mathcal{O}(\sqrt{N})$. The density of states of $S(t)$ can be made equal to those of $M(t)$ by scaling the latter by $1/\sqrt{2\pi}$. We therefore expect the spectral statistics of the RP for a given $\epsilon$ at the middle of the spectrum to match those of the CRP with $\epsilon \to \epsilon/\sqrt{2\pi}$. We remark that $M(t)$ has a density of states that remains constant during the Brownian motion process for $\gamma > 1$ [30].

The eigenstates $|\psi_n\rangle$ of $S(t)$ associated with the eigenvalues $e^{i\theta_n(t)}$ evolve stochastically according to

$$d|\psi_n(t)\rangle = \sqrt{dt} \sum_{m \neq n} \frac{X_{mn}}{2\tan\frac{1}{2}[\theta_n(t) - \theta_m(t)]} |\psi_m(t)\rangle - \frac{dt}{2} \sum_{m \neq n} \frac{(X_{mn})^2}{4\tan^2\frac{1}{2}[\theta_n(t) - \theta_m(t)]} |\psi_n(t)\rangle,$$
(12)

where again $d|\psi_n(t)\rangle = |\psi_n(t+dt)\rangle - |\psi_n(t)\rangle$. In line with the arguments for the eigenvalues, one might expect that the RP and CRP have similar eigenstate statistics. We remark that no explicit construction for the matrices resulting from circular Dyson Brownian motion processes is known [63]. Moreover, we note that the CRP can not be constructed as $S = e^{iH}$ for $H$ sampled from the RP ensemble. Indeed, for example then the density of states of $S$ would not be uniform.

## 4 Numerical evaluation

The results below are obtained through a numerical evaluation of Eq. (9) up to $t = \epsilon^2/N^\gamma$ using the algorithm described Ref. [52] (Sec. 11.2.1). This algorithm utilizes a different and numerically equivalent expression for of the stochastic equations (9), given by

$$S(t + dt) = S(t)e^{i\sqrt{dt}X}.$$
(13)

This description has the computational advantages that no decomposition $S = U^T U$ has to be made, and that $S(t)$ remains unitary even for finite $dt$. In the numerical evaluations, we average over at least $10^6$ eigenvalues or eigenstates. We take $\epsilon = 1$ and $10^4$ time steps to reach $t = \epsilon^2/N^\gamma$. Empirically, this time step turns out to be small enough for convergence of the results. We have validated that our results are visibly indistinguishable from results obtained when taking the number of time steps half as large, and found that no qualitative differences can be observed when taking the number of time steps an order of magnitude smaller. The numerical evaluation of Eq. (9) requires diagonalization or matrix-matrix multiplications in each step, making it is computationally expensive. We therefore restricted to matrix dimensions $N \leq 1000$, which still provide sufficient numerical evidence for the claims of this work.

We denote the eigenphases for samples from the CRP ensemble by $\theta_n$, which we label in ascending order ($\theta_{n+1} \geq \theta_n$). First, we consider the average ratio of consecutive level spacings $\bar{r}$. The ratios $r_n$ of consecutive level spacings are defined in Eq. (2), with $E_n$ replaced by the phases $\theta_n$. The left panel of Fig. 1 shows $\bar{r}$ as a function of $\gamma$ for several matrix dimensions. As discussed in Sec. 2, the RP ensemble is characterized by $\bar{r} \approx 0.530$ for $\gamma < 2$ (Wigner-Dyson) and $\bar{r} \approx 0.386$ (Poissonian) for $\gamma \geq 2$ in the large-$N$ limit, which is consistent with our results. The level statistics exhibit a phase transition with critical exponent $\nu = 1$ from Wigner-Dyson

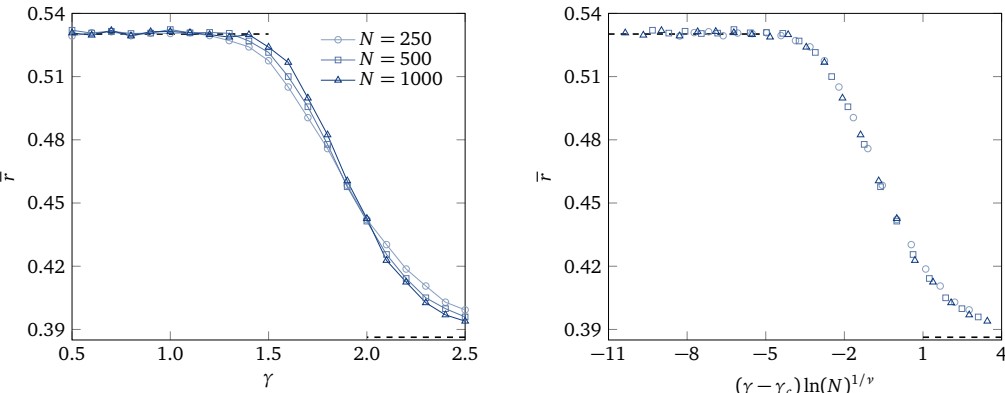

Figure 1: The average ratio of consecutive level spacing spacings $\overline{r}$ as a function of $\gamma$ (left panel) and $(\gamma - \gamma_c)\ln(N)^{1/\nu}$ with $\gamma_c = 2$ and $\nu = 1$ (right panel) for matrix dimensions $N = 250$, $500$, and $1000$. Poissonian and Wigner-Dyson level statistics are characterized by respectively $\overline{r} \approx 0.386$ and $\overline{r} \approx 0.530$ (dashed lines).

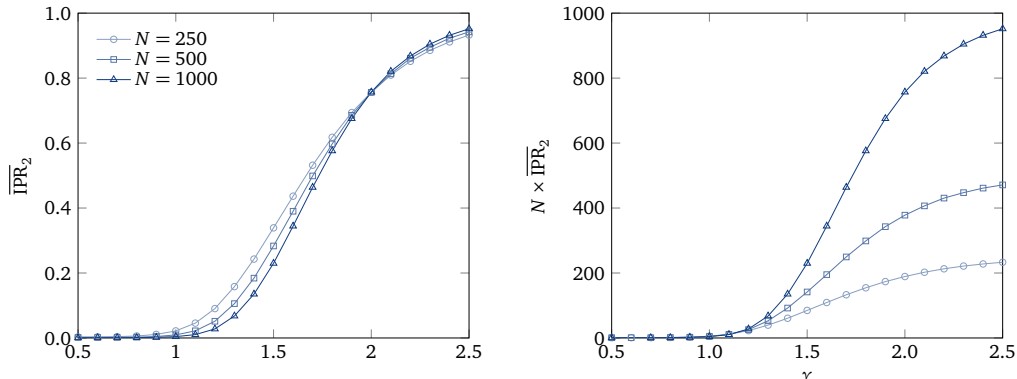

Figure 2: The average of the inverse participation ratio $\mathrm{IPR}_2$ (left panel) and the value scaled by $N$ (right panel) as a function of $\gamma$ for matrix dimensions $N = 250$, $500$, and $1000$.

to Poissonian at $\gamma = \gamma_c = 2$. For the RP ensemble, a scaling collapse of $\overline{r}$ as a function of $(\gamma - \gamma_c)\ln(N)^{1/\nu}$ has been observed in Ref. [33]. The right panel shows that this observation can be made as well for the CRP ensemble.

Next, we consider eigenstates of the CRP ensemble. We focus on the basis in which $S(0)$ is diagonal. We denote eigenstates of $S$ and $S_0$ by respectively $|\psi_n\rangle$ and $|n\rangle$, with the eigenvalue $e^{i\theta_n}$ corresponding to $|\psi_n\rangle$. We first focus on the eigenstate inverse participation ratio $\mathrm{IPR}_2$ as given in Eq. (3). Asymptotically in $N$, the RP ensemble is characterized by $\mathrm{IPR}_2 \sim N^{-1}$ for $\gamma \leq 1$, and $\mathrm{IPR}_2 = \mathcal{O}(1)$ for $\gamma \geq 2$ as discussed in Sec. 2. In the intermediate region $1 < \gamma < 2$, the eigenstates are fractal, and are characterized by $\mathrm{IPR}_2 \sim N^{-(2-\gamma)}$. Fig. 2 shows the average $\overline{\mathrm{IPR}_2}$ (left panel) and $N \times \overline{\mathrm{IPR}_2}$ (right panel) as a function of $\gamma$, for the same matrix dimensions as studied above. The results we obtain for the CRP ensemble are again consistent with the large-$N$ behavior for the RP ensemble.

Finally, we consider the shape of the eigenstates. The eigenstates of the RP ensemble obey Breit-Wigner statistics [30]. For the CRP ensemble we aim to show [cf. Eq. (4)] that

$$\overline{|\langle m|\psi_n\rangle|^2} \sim \frac{1}{\left(\theta_n - \theta_m^{(0)}\right)^2 + \Gamma^2}, \tag{14}$$

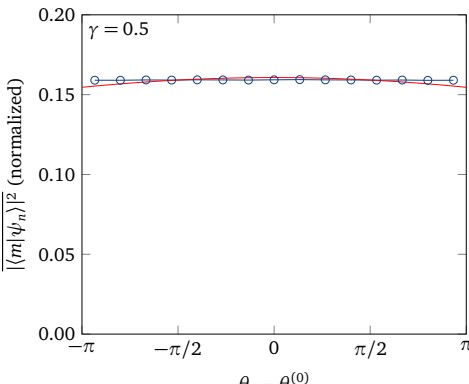
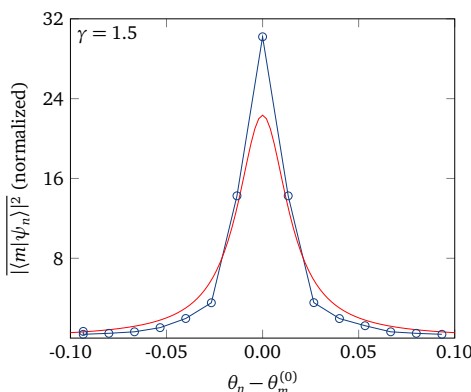

Figure 3: The average of $|\langle m|\psi_n\rangle|^2$ as a function of $\theta_n - \theta_m^{(0)}$ (markers), combined with the evaluation of Eq. (14) with $\rho = 1000/(2\pi)$ (red solid lines) for $N = 1000$ with $\gamma = 0.5$ (left panel) and $\gamma = 1.5$ (right panel). Each of the curves is normalized to unit area over the plotted range. Blue lines connecting the markers serve as a guide to the eye.

where $\theta_m^{(0)} = \theta_m(0)$ as introduced in Eq. (10). Fig. 3 shows the average $\overline{\langle m|\psi_n\rangle|^2}$ as a function of $\theta_n - \theta_m^{(0)}$ for $\gamma = 0.5$ (left panel) and $\gamma = 1.5$ (right panel) for the largest matrix dimension $N = 1000$. The smooth curves show the evaluation of Eq. (14) with $\Gamma$ obtained from Eq. (5), and the density of states is given by $\rho = 1000/(2\pi) \approx 159.2$.

For $\gamma = 0.5$ the expected spreading width has a value $\Gamma \approx 15.8$, such that the resulting Lorentzian shape has a width exceeding the width $2\pi$ of the spectrum, leading to a slight mismatch with the observed results. For $\gamma = 1.5$ the spreading width evaluates to $\Gamma \approx 1.6 \times 10^{-3}$, which is comparable to the mean level spacing $\rho^{-1} \approx 6.3 \times 10^{-4}$. Eq. (4) relies on a continuum approximation for the density of states [30]. Presumably due to this, the agreement between our data and the expected results is not perfect, although qualitative agreement, and a pronounced difference between the structures of the eigenstates for $\gamma = 0.5$ and $\gamma = 1.5$ is clearly seen. To our knowledge, this characteristic of eigenstates has not been previously investigated for unitary matrix ensembles.

## 5 Conclusions and outlook

We have proposed a unitary (circular) analogue of the Rosenzweig-Porter (RP) ensemble, defined as the outcome of a Dyson Brownian motion process. We have numerically verified that some key statistics of both the eigenvalues and the eigenstates of the circular analogue match the behaviour of the RP ensemble. The circular analogue, therefore, can serve as a phenomenological model for the level statistics and fractality of eigenstates as observed across the many-body localization transition for periodically driven systems.

Motivated by, among other observations, the suggestion that eigenstates near the many-body localization transition are multifractal [64–66], several generalizations of the RP ensemble have been proposed [29, 43–46]. It would be interesting to see if similar generalization could be constructed for the circular RP ensemble, next to other Floquet models with multifractal eigenstates [47–50]. This might for example be achievable by considering stochastic processes with correlated increments, which has been initialized for the RP ensemble recently [67]. Second, the circular RP ensemble could potentially be of value in studies on random quantum circuits [68] as a non-maximally random building block, analog to e.g. the proposal made in Ref. [69].

# Acknowledgements

We thank Ivan M. Khaymovich for the careful reading of the manuscript and providing us with many useful comments, as well as Lea F. Santos for useful discussions.

**Funding information** YB acknowledges support from the Israel Science Foundation (grants No. 218/19 and 527/19). WB acknowledges support from the Kreitman School of Advanced Graduate Studies at Ben-Gurion University.

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
