# Peer review of "Circular Rosenzweig-Porter random matrix ensemble"

_SciPost Physics, doi:SciPost Phys. 12, 082 (2022)_

## Round 1 · Referee Report · Ivan Khaymovich (Referee 1) · 2021-11-30

Strengths

1 - Generalization of a Rosenzweig-Porter model and its properties from Hermitian (or real symmetric) to circular (or unitary) case.
2 - The model is new and instructive.
3 - Extensive numerical data support the analysis.
4 - The numerics is supported by Dyson-Brownian motion equations
5 - The paper is very clearly written.
6 - Writing is well paced, and with an explanatory style.

Weaknesses

1 - Numerically complicated realization of a circular (unitary) variant of a Rosenzweig-Porter model (see the report).
2 - Equivalence of Dyson Brownian motion equations to the Hermitian case is not completely clear.

Report

In this paper the authors define a model which generalize the Rosenzweig-Porter (RP) model to the circular (unitary) case.

They present analytic derivation of the Dyson Brownian motion equations for the model, connect them to the ones of the Hermitian RP model, and provide some numerical evidence that shows agreement with these approximate equivalence. In particular, authors explore ratio statistics of adjacent energy levels and the inverse participation ratio showing the same non-ergodic behavior at $1<\gamma<2$, with fractal wave functions and fractal dimensions, $D_{q>1/2} = 2-\gamma$, similar to the Hermitian RP case.

In addition, the authors introduce a useful tool to probe fractality, namely, the wave-function profile versus the diagonal energy $\theta_n$, averaged over the off-diagonal element.

The manuscript is well paced, and adopts a useful pedagogic style. However, despite this, there are several things which should be discussed:

1) It seems that the Dyson-Brownian motion process of Eq. (9) is overcomplicated. One can get rid of decomposing the matrix $S = U^T U$ at each step by considering either of three following variants: a. $S(t+dt) = S(t) e^{i X \sqrt{dt}}$ b. $S(t+dt) = e^{i X \sqrt{dt}}S(t)$ c. $S(t+dt) = e^{i X \sqrt{dt}/2}S(t) e^{i X \sqrt{dt}/2}$ All of them provide the same equation for the eigenphases (11), while the equations for eigenstates are very close to (12) leading to a. $d|\psi_n^{(a)}(t)\rangle = \sqrt{dt} \sum_{m\ne n} \frac{X_{mn}}{2 \sin\tfrac12[\theta_n (t)- \theta_m(t)]}|\psi_m^{(a)}(t)\rangle - \sqrt{dt} \frac{dt}{2} \frac{|X_{mn}|^2}{4 \sin^2\tfrac12[\theta_n (t)- \theta_m(t)]}|\psi_n^{(a)}(t)\rangle$ b. $|\psi_n^{(b)}(t)\rangle = e^{i \theta_n(t)}|\psi_n^{(a)}(t)\rangle$ c. $|\psi_n^{(c)}(t)\rangle = |\psi_n(t)\rangle$

2) It would be useful to show the extracted fractal dimensions $D_q$ instead of collapses of the inverse participation ratios in Fig. 2.

However, my concerns about the manuscript are easily addressed, and overall I feel the results represent a useful addition to the literature. As a result recommend the manuscript for publication after minor revision (see Requested changes).

Requested changes

Minor changes:

1 - In the last paragraph of page 2 please clarify from the very beginning that you consider the level statistics of consecutive level spacing. Otherwise one should take into account the fact that in the Rosenzweig-Porter model at $1<\gamma<2$ the level statistics is of Wigner-Dyson form only until a certain energy scale $\Gamma \sim N^{1-\gamma}$, while beyond it it is of Poisson kind (see [23, 34]).

2 - Before Eq. (7) it is better to mention that the perturbation theory is developed over a small parameter $dt$.

3 - In the expression for $d|\psi_n(t)\rangle$ after Eq. (8) the vertical line is missing in r.h.s.

4 - After Eq. (9) it is better to provide a check of unitarity of the matrix or write the Dyson Brownian motion in the unitary preserving form $S(t+dt) = U^T e^{i X \sqrt{dt}} U$.

5 - It seems that there is misprint in the numerator of the last fraction of Eq. (12): it should be (like in Eqs. (8, 11)) $(X_{mn})^2$ instead of $(X^2)_{mn}$.

6 - It would be also useful to describe in the text more on the parameters of the numerical simulations, like the size of the step $dt$ used, and explain - why the author choose a certain amount of step in the Dyson-Brownian motion ($10^4$).

7 - It seems that the authors assume $\theta_n<\theta_{n+1}$ to be sorted in order to plot Fig. 3. Please clarify it in the text.

8 - The notion of the first-order phase transition is misused in the second paragraph of Sec. 4: the critical exponent $\nu = 1$ still corresponds to the continuous transition, which is of the second order, while the first-order transition should be characterized by - a finite jump in the order parameter, - hysteretic behavior (overcooled/overheated stated), - metastability and so on. Please correct this.

9 - There is no top or bottom panel in Figs. 2 and 3 described - in the second to the last paragraph in page 5 - and in the first paragraph of page 7, respectively. Please either use left and right ones or go to the labels like (a) and (b).

10 - References: - Please supplement the motivation to quantum random circuit applications [68] mentioned in the conclusions and outlook section by an earlier relevant paper by the same authors (https://journals.aps.org/prx/abstract/10.1103/PhysRevX.8.041019). - Please correct the spelling of a surname of Lev B. Ioffe in [43]. - Please correct the spelling of ‪François Huveneers in [38].

  • validity: high
  • significance: high
  • originality: high
  • clarity: high
  • formatting: excellent
  • grammar: excellent

Author:  Wouter Buijsman  on 2021-12-23  [id 2044]

(in reply to Report 1 by Ivan Khaymovich on 2021-11-30)

We are grateful to the referee for the careful reading and the positive evaluation of our manuscript. Below, we reply to the comments in the order in which they appear in the report.

Referee: It seems that the Dyson-Brownian motion process of Eq. (9) is overcomplicated. One can get rid of decomposing the matrix $S = U^T U$ at each step.

It is correct that the dynamical equations can be obtained without the decomposition of the matrix $S$. In fact, variant (a) of the alternative constructions proposed by the Referee is what we use in the numerical evaluation. Aiming to keep our notation consistent with the pioneering work [Dyson62], we decided to keep the formulation of the Dyson Brownian motion process in its current form. Instead, we have extended the description of the numerical algorithm, now including an explicit remark on this simplification.

Referee: It would be useful to show the extracted fractal dimensions $D_q$ instead of collapses of the inverse participation ratios in Fig. 2.

It is challenging to extract reliable conclusions on the fractal dimensions of the eigenstates given the matrix dimensions we can access numerically. We have attached a plot of the extracted fractal dimension $D_2$ as a function of gamma, which is obtained as the slope of a least-square fit $\log(\overline{\text{IPR}}_2) = -D_2 \log(N) + X$ where $X$ is an offset. A plot showing the data (dots) and the fits (dashed lines) is attached as well. In addition to the data shown in the manuscript, we have included data for matrix dimensions $N=350$ and $N=700$ in the analysis. From the observation that the fractal dimension acquires values above one and below zero, we conclude that data for significantly larger system sizes is required to properly draw such a plot. As such, we have decided not to include it in the manuscript.

Referee: In the last paragraph of page 2 please clarify from the very beginning that you consider the level statistics of consecutive level spacing. Otherwise one should take into account the fact that in the Rosenzweig-Porter model at $1 < \gamma < 2$ the level statistics is of Wigner-Dyson form only until a certain energy scale $\Gamma \sim N^{1-\gamma}$, while beyond it it is of Poisson kind (see [23, 34]).

We have clarified on this.

Referee: Before Eq. (7) it is better to mention that the perturbation theory is developed over a small parameter $dt$.

We have followed this suggestion.

Referee: In the expression for $d | \psi \rangle$ after Eq. (8) the vertical line is missing in r.h.s.

We have corrected this misprint.

Referee: After Eq. (9) it is better to provide a check of unitarity of the matrix or write the Dyson Brownian motion in the unitary preserving form $S(t+dt) = U^T e^{i \sqrt{dt} X} U$.

We have added a remark on the persistence of unitarity for small dt.

Referee: It seems that there is misprint in the numerator of the last fraction of Eq. (12): it should be (like in Eqs. (8, 11)) $(X_{mn})^2$ instead of $(X^2)_{mn}$.

We have corrected this misprint

Referee: It would be also useful to describe in the text more on the parameters of the numerical simulations, like the size of the step $dt$ used, and explain why the author choose a certain amount of step in the Dyson-Brownian motion ($10^4$).

We have elaborated on this.

Referee: It seems that the authors assume $\theta_n < \theta_{n+1}$ to be sorted in order to plot Fig. 3. Please clarify it in the text.

This is correct. We have clarified this.

Referee: The notion of the first-order phase transition is misused in the second paragraph of Sec. 4.

We have corrected this.

Referee: There is no top or bottom panel in Figs. 2 and 3.

We have corrected these misprints.

Referee: Please supplement the motivation to quantum random circuit applications [68] mentioned in the conclusions and outlook section by an earlier relevant paper by the same authors (https://journals.aps.org/prx/abstract/10.1103/PhysRevX.8.041019). Please correct the spelling of a surname of Lev B. Ioffe in [43]. Please correct the spelling of ‪François Huveneers in [38].

We have included this reference, and corrected the misprints.

Attachment:

attachment.pdf

---

## Round 1 · Referee Report · Vladimir Kravtsov (Referee 2) · 2021-12-2

Strengths

  1. The subject is important, as the Rosenzweig-Porter model is one of a few proven models which supports the extended non-ergodic phase in the thermodynamic limit 2.The unitary analog of the Hamiltonian representation of the Rosenzweig-porter has a constant density of eigenvalues and does not need unfolding procedure at any scale

Weaknesses

  1. Lack of an analytical expression for the measure of the unitary matrices of the circular Rosenzweig-Porter ensemble
  2. Expensive numerical procedure of generating the ensemble

Report

I think that this paper is important and timely, since the Rosenzweig-Porter ensemble is rigorously shown to possess a (mono)-fractal eigenstates in an extended interval of parameters, and not only in certain critical points. The unitary representation of this ensemble (the circular Rosenzweig-Porter ensemble) is a step forward with important potential applications in quantum computing. The authors suggest a stochastic procedure of generating the ensemble of unitary random matrices which eigenvalues exp[-i\phi] have a constant mean density of phases \phi which exhibit the statistical properties similar to those in the Hamiltonian representation of the Rosenzweig-Porter ensemble. In particular, their r-statistics shows a transition between the ergodic and non-ergodic extended phase in addition to the localization transition between the non-ergodic extended and localized phase. The same transitions exhibit themselves in the eigenfunction inverse participation ratio.
An importance of the circular ensemble is that it is an ensemble of unitary random matrices which can be incorporated into the quantum computing schemes. Another important property is the constant mean density of phases which allows to avoid the unfolding procedure at all scales.
The paper is clearly written and contains convincing numerical evidence of the existence of three distinctly different phases.
I would recommend this paper to be published in its present form. The only clarification needed in my opinion is the procedure of decomposition of the S matrix in terms of the unitary ones.

Requested changes

None

  • validity: high
  • significance: high
  • originality: good
  • clarity: good
  • formatting: excellent
  • grammar: excellent

Author:  Wouter Buijsman  on 2021-12-23  [id 2045]

(in reply to Report 2 by Vladimir Kravtsov on 2021-12-02)

We would like to thank the referee for his very positive assessment of our manuscript. Please see below for our reply.

Referee: The only clarification needed in my opinion is the procedure of decomposition of the S matrix in terms of the unitary ones.

We have elaborated on this in the revised version of our manuscript. Besides, we have made it more explicit that this decomposition is not actually required in performing numerical simulations (see also the report by dr. I. Khaymovich).

---

## Round 1 · Referee Report · Anonymous (Referee 3) · 2021-12-7

Strengths

1 - clearly written
2 - well referenced
3 - fills a gap in the literature

Weaknesses

1 - the proposed ensemble does not offer any clear advantage nor different predictions from the standard (hermitian) one
2 - none of the expectations (necessary acceptance criteria) is met here

Report

In this work, the authors propose a ‘circular’ (unitary) version of the classical Rosenzweig-Porter random matrix model, and study its statistical properties (mainly numerically).
After a concise but essentially complete introduction, section 2 is devoted to a nice and well-referenced summary of the ‘classical’ (real symmetric) version of the R-P ensemble.
In section 3, the unitary model is built using a very standard construction by Dyson, dating back to 1962. The only ‘new’ aspect here is the identification of the time variable as eps^2 / N^gamma . Moreover, both an analytical argument provided at the end of section 3, and the numerical simulations in section 4 (for spacing ratios, IPR, shape of eigenstates) confirm that the newly proposed unitary ensemble is not statistically different (at least on ‘microscopic’ scales) from the standard R-P ensemble. This is not very surprising to a trained RMT eye, and in some sense makes it unclear (at least to me) what the innovative contribution would be here. This paper nominally fills a gap - but the ‘new’ ensemble does not seem to offer any significant advantage or different predictions with respect to the standard version of it. Maybe some better/superior insight could come from the ‘outlook’ (material for further research), but at the moment I am unsure that the paper meet the required standard for being published in SciPost, as it certainly does not [Detail a groundbreaking theoretical/experimental/computational discovery]; nor [Present a breakthrough on a previously-identified and long-standing research stumbling block]; nor [Provide a novel and synergetic link between different research areas]. And it is difficult to argue that it would [Open a new pathway in an existing or a new research direction, with clear potential for multipronged follow-up work] in the absence of further strong evidence that the ‘unitary’ version could do/say something substantially different from the ‘hermitian’ version.
  • validity: ok
  • significance: low
  • originality: low
  • clarity: high
  • formatting: excellent
  • grammar: excellent

Author:  Wouter Buijsman  on 2021-12-23  [id 2046]

(in reply to Report 3 on 2021-12-07)

We thank the referee for reviewing our manuscript. Please see below for our reply.

Referee: In section 3, the unitary model is built using a very standard construction by Dyson, dating back to 1962. The only ‘new’ aspect here is the identification of the time variable as eps^2 / N^gamma. Moreover, both an analytical argument provided at the end of section 3, and the numerical simulations in section 4 (for spacing ratios, IPR, shape of eigenstates) confirm that the newly proposed unitary ensemble is not statistically different (at least on ‘microscopic’ scales) from the standard R-P ensemble. This is not very surprising to a trained RMT eye, and in some sense makes it unclear (at least to me) what the innovative contribution would be here.

Dyson Brownian motion is concerned with statistics of eigenvalues only. As such, it does not make statements about the statistics of eigenstates. The correspondence between the statistics of eigenstates for the Rosenzweig-Porter ensemble and the circular analogue is to the best of our knowledge not established in the literature. We thus believe that an investigation like the one presented in sections 3 and 4 of our manuscript is pertinent.

Referee: This paper nominally fills a gap - but the ‘new’ ensemble does not seem to offer any significant advantage or different predictions with respect to the standard version of it.

Our primary aim is to use tools from random matrix theory to establish a potentially physically relevant random matrix ensemble. As mentioned in the referee report by prof. Kravtsov, the Rosenzweig-Porter ensemble is one of the few ensembles characterized by a domain of parameter values for which the eigenstates are fractal. Therefore, a circular analogue is potentially relevant for periodically driven many-body localized systems (section 5),quantum computing (referee report by prof. Kravtsov), or random unitary circuits (section 5).

While periodically driven and static many-body localized systems share similar features, they have many different properties. Examples include the dynamical behavior in the ergodic phase and the existence of a mobility edge (see e.g. DOI:0.1016/j.physrep.2020.03.003 for a review.). Therefore, the circular analogue of the Rosenzweig-Porter ensemble that we present here can shed light on the differences and similarities of the physical systems, in particular close to the many-body localization transition.

---

## Round 2 · Referee Report · Anonymous (Referee 3) · 2021-12-23

Report

I believe the authors have satisfactorily allayed my previous concerns, and have improved their work following the thorough comments of the other reviewers.

---

## Round 2 · Referee Report · Vladimir Kravtsov (Referee 2) · 2021-12-23

Report

I am fully satisfied by the amendments recommended my me. The paper can by published.

---

## Round 2 · Referee Report · Ivan Khaymovich (Referee 1) · 2021-12-23

Report

In the revised version of the manuscript the authors carefully addressed all the comments of all three referees and now the manuscript is ready to be published in SciPost Physics.

---

## Round 2 · List of Changes

Section 2 - Elaborated on the notion that we are studying level statistics on a microscopic scale.

Section 3 - Clarified on the unitarity of $S(t)$.

Section 4

  • Elaborated more on the algorithm used to generate numerical data.

  • Corrected a misprint on the interpretation of the finite-size scaling collapse.

Section 5 - Added Ref. [68] (resubmission).

References - Fixed misprints mentioned in the reports.

  • Updated Ref. [24] (resubmission) from the preprint to the published version.

  • Updated Ref. [67] (resubmission) from the preprint to the published version.

---

## Editorial Decision

published